# Evaluation of CDK9 Inhibition by Dinaciclib in Combination with Apoptosis Modulating izTRAIL for the Treatment of Colorectal Cancer

**DOI:** 10.3390/biomedicines11030928

**Published:** 2023-03-16

**Authors:** Xiao Shen, Anna-Laura Kretz, Sandra Schneider, Uwe Knippschild, Doris Henne-Bruns, Marko Kornmann, Johannes Lemke, Benno Traub

**Affiliations:** Department of General and Visceral Surgery, Ulm University Hospital, Albert-Einstein-Allee 23, 89081 Ulm, Germany

**Keywords:** TNF-related apoptosis-inducing ligand (TRAIL), cyclin-dependent kinases (CDKs), CDK9, dinaciclib, colorectal cancer (CRC), drug target

## Abstract

Treatment options for colorectal cancer (CRC), especially in advanced stages are still insufficient. There, the discovery of Tumor necrosis factor (TNF)-related apoptosis-inducing ligand (TRAIL) was a bright spot. However, most cancers show resistance toward apoptotic signals. Cyclin-dependent kinase 9 (CDK9) plays a crucial role in cell cycle progression in most tissues. We recently demonstrated the role of CDK9 in mediating TRAIL resistance. In this work, we investigated the role of CDK9 in colorectal cancer. Immunohistochemical analysis of CDK9 expression in cancer and normal tissues of CRC specimens was performed. The effect of selective CDK9 inhibition in combination with TRAIL on CRC cells was analyzed via cell viability, colony formation, and induction of apoptosis by flow cytometry. The mechanism of action was conducted via western blotting. We now have confirmed overexpression of CDK9 in cancer tissues, with low expression associated with poorer survival in a subset of CRC patients. In-vitro, CDK9 inhibition could strongly promote TRAIL-induced cell death in TRAIL-resistant CRC cells. Mechanistically, CDK9 inhibition induced apoptosis by downregulation of antiapoptotic proteins, myeloid leukemia cell differentiation protein 1 (Mcl-1) and FLICE-inhibitory protein (c-FLIP). Overall, we identified CDK9 as a prognostic marker and combined CDK9 inhibition and TRAIL as a novel and promising therapeutic approaches for colorectal cancer.

## 1. Introduction

Colorectal cancer (CRC) with more than 1.9 million new cases being diagnosed and 935,000 deaths being estimated to occur in 2020 is still the third most commonly diagnosed cancer and the second leading cause of cancer-related death globally, exceeded only by lung cancer [1]. The five-year survival rate for colorectal cancer is less than 60% in European countries [2]. CRC can be treated or even cured surgically [3], mainly depending on the cancer stage, mutated genes [4], and individual conditions. Unfortunately, colorectal cancer can harbor resistance to conventional chemo- or radio-therapy. Thus, it is critically important that better treatment strategies need to be sought.

Tumor necrosis factor (TNF)-related apoptosis-inducing ligand (TRAIL) was first described in the 1990s [5]. It could selectively induce apoptosis in cancer cells in vitro and in vivo [5,6,7]. TRAIL acting as an extracellular signal triggers apoptosis via binding to membrane-bound death receptors 4/5 (DR4 and DR5), which transmit death signals through a cytoplasmic “death domain” motif, thereby inducing the formation of the pro-apoptotic death-inducing signaling complex (DISC) and the downstream activation of the apoptotic cascade to execute cell apoptosis [8,9]. In order to exploit those features, TRAIL-receptor agonists (TRAs) were developed. Unfortunately, most TRAs failed in clinical usage due to the intrinsic insensitivity of cancer cells to TRAIL [10]. Consequently, future TRAIL-based therapies will require the addition of TRAIL-sensitizing agents.

Cyclin-dependent kinases (CDKs) act as a crucial regulator of cell cycle progression and transcription. The deregulation of CDKs and CDK-mediated pathways are attributed to tumorigenesis and the progression of different human cancer entities [11,12,13,14]. Among them, cyclin-dependent kinase 9 (CDK9), together with cyclin-T1, constitutes the positive transcription elongation factor (pTEFb), which critically regulates transcriptional elongation by phosphorylation of RNA-Polymerase II (RNA Pol II) [15,16]. Suppression of RNA Pol II by CDK9 inhibition has been shown to block transcription elongation leading to suppression of short-living anti-apoptotic proteins, such as Myeloid leukemia cell differentiation protein 1 (Mcl-1), Survivin, and X-linked inhibitor of apoptosis protein (XIAP), thereby promoting apoptosis [17]. Previous studies from our research team have confirmed that CDK9 is highly expressed in pancreatic cancer tissues [18]. Furthermore, selective CDK9 inhibition overcame TRAIL resistance by the suppression of anti-apoptotic proteins Fas-associating protein with a novel death domain (FADD)- like IL-1-converting enzyme (FLICE)-inhibitory protein (c-FLIP) and Mcl-1 in non-small cell lung cancer (NSCLC), pancreatic, ovarian and liver cancers [19]. Based on the high potency of CDK9 inhibition as a cancer cell-selective TRAIL-sensitizing strategy, we believe this is a potent new anti-cancer strategy. Consequently, more attention has been paid to the development and evaluation of selective CDK9 inhibitors in order to target the deregulated cell cycle progression of cancer cells [12,14].

Several CDK9 inhibitors undergoing clinical testing have been identified. Dinaciclib (SCH727965) appears to selectively inhibit CDK1, 2, 5, and 9 [20] and is currently under clinical investigation, for example in metastatic triple-negative breast cancer and chronic lymphocytic leukemia [21,22,23]. The activity of Dinaciclib has been verified in various animal models [24].

In this study, we investigated the role of CDK9 in colorectal cancer as a prognostic marker and its potential via combination with TRAIL agent as a novel therapeutic target. CDK9 was highly expressed in colorectal cancer tissues and patients with high expression showed a tendency towards prolonged survival. Moreover, CDK9 inhibition by Dinaciclib resulted in markedly decreased colorectal cancer cell viability and profoundly enhanced TRAIL-mediated apoptosis of resistant colorectal cancer cells via suppression of short-living anti-apoptotic proteins Mcl-1 and c-FLIP. Thus, CDK9 inhibition in combination with TRAIL is a novel, promising therapeutic strategy for colorectal cancer.

## 2. Materials and Methods

### 2.1. Patients and Tissue Samples

We collected tissue specimen of patients diagnosed with colorectal cancer (CRC) that were treated in the Department of General and Visceral Surgery of the University Hospital of Ulm between November 2003 and October 2014. Patients who had received preoperative chemotherapy were excluded. Our study includes cancer and adjacent normal tissues from 175 colorectal cancer patients. Samples were collected immediately during surgery. Informed consent was obtained from all patients. Histopathological and clinical data including patients’ age, sex, cancer differentiation (grading), T classification, lymph node invasion, distant metastases, and cancer stage (according to the Union for International Cancer Control (UICC)) [25] were obtained (Table 1). The study was performed with permission of the independent local ethics committee of the University of Ulm (approvals 112/2003, and 268/2008, and 235/2015).

### 2.2. Cell Lines

The human colorectal cancer cell lines HT-29, HCT-116, SW-480, SW620, and DLD-1 were purchased from the American Type Culture Collection (ATCC, Manassas, VA, USA). HT-29 and HCT-116 were cultured in McCoy’s 5A Medium (Gibco^®^, Thermo Fisher Scientific, Waltham, MA, USA). SW-480, SW-620, and DLD-1 were cultured in RPMI Medium 1640 (Gibco^®^, Thermo Fisher Scientific, Waltham, MA, USA). The media were supplemented with 10% fetal bovine serum (FBS) (Biochrom AG, Merck Millipore, Darmstadt, Germany), 1% L-glutamine (GE Healthcare, Amersham, Buckinghamshire, UK), and 1% penicillin/streptomycin (Thermo Fisher Scientific, Waltham, MA, USA).

### 2.3. Immunohistochemistry

Formalin-fixed paraffin-embedded sections of normal and cancer tissues were used for immunohistochemical analysis. Hematoxylin-eosin (HE)-stained specimens were used for evaluating tumor burden and tissue quality. Sections of 1 µm thickness were deparaffinized and rehydrated in xylene followed by rehydration via transfer through graded alcohols. The slices were heated in the microwave using a pressure cooker. Microwave setup was 450 W for 15 min followed by 270 W for 10 min. CDK9 (C12F7) antibody was diluted 1:150 (Cell Signaling Technology, Danvers, MA, USA). Evaluation of immunohistochemical staining was performed by one person. CDK9 expression score was calculated as follows: Staining intensity was evaluated (0 = no staining, 1(+) = low intensity, 2(++) = moderate intensity, 3(+++) = high intensity) of 500 carcinoma cells of each cancer tissue slide and 500 epithelial cells in each normal-tissue slide. The number of cells of each staining intensity was multiplied by the intensity value. Thus, the score ranged from 0 to 1500. The CDK9 score was generated for both normal tissue and carcinoma tissue.

### 2.4. Cell Viability Assay

Cell viability in this study was determined by conventional 3-(4,5-dimethy-thiazol-2-yl)-2,5-diphenyl tetrazolium bromide (MTT) assay as described previously [26]. Combination therapy treatment was performed by co-incubating HT-29, HCT-116, SW-480, SW-620, and DLD-1 cells with isoleucine-zipper-tagged TRAIL (izTRAIL) kindly provided by Henning Walczak (UCL Cancer Institute London, UK, now Institute for Biochemistry I, Centre for Biochemistry, University of Cologne, Germany), and Dinaciclib (ChemieTek Company, Indianapolis, IN, USA) at different concentrations for 24, 48 and 72 h. Every assay was repeated more than three times.

### 2.5. Analysis of Long-Term Survival and Colony Formation

Cells were seeded in 6-well plates and incubated with dimethyl sulfoxide (DMSO), izTRAIL, Dinacicllib, or a combination of the latter for 24 h. After incubation for 7 days, cells were fixed in 4% Paraformaldehyde (PFA) (Serva, Heidelberg, Germany) for 15 min at room temperature, and stained with crystal violet (Sigma-Aldrich^®^, St. Louis, MI, USA; 250 mg crystal violet powder in 250 mL 95% methanol)

Colony formation assay was performed as described previously [27] according to the protocol of Millipore (Catalog No. ECM570, Merck KGaA, Darmstadt). Respective treatments were added to the top agar. The cells were incubated for 28 days until colonies were formed. Formed colonies were photographed for three times in a bright field at specific positions located on each fan-shaped center of every well divided into three equal parts and subsequentially visualized and quantified using MTT staining. The colony count was performed using ImageJ (National Institute of Health, Bethesda, MD, USA).

### 2.6. Analysis of Cell Cycle and Apoptosis by Flow Cytometry

When dead cells were stained with PI and analyzed by flow cytometry, they displayed a broad hypodiploid (sub G1) peak, which can be easily discriminated from the narrow peak of cells with normal (diploid) deoxyribonucleic acid (DNA) content in the red fluorescence channels [28]. Cell cycle analysis was performed using the BD Cycletest™ Plus DNA Reagent Kit (BD Biosciences, San Jose, CA, USA) as described previously [29]. Cells seeded into 60 mm-dishes were treated by DMSO, 10 ng/mL izTRAIL, 25 nM Dinaciclib, or the combination of Dinaciclib and izTRAIL for 24 h. After harvesting and centrifugation, the cells were resuspended in a PI staining solution (Sigma Life Science, USA). Cell cycle profiles were obtained using MACS Quant flow cytometer (Miltenyi Biotec, Bergisch Gladbach, Germany) and FlowJo software (Becton Dickinson Biosciences, San Jose, CA, USA).

### 2.7. Western Blot Analysis

Western blot was performed as described previously [30]. Antibody concentrations of primary antibodies were used as follows: Anti-DR4 (1:1000; Pro Sci, Fort Collins, CO, USA), anti-caspase-8 (1:1000; Enzo), anti-CDK9, anti-PARP, anti-FADD, anti-Bax, anti-Bak, anti-Mcl-1, anti-Bcl-xL, anti-Bcl-2, anti-cFLIP (AdipoGen, San Diego, CA, USA), anti-cIAP1, anti-cIAP2, anti-XIAP, anti-survivin (1:1000; Cell Signaling Technology), anti-caspase-9 (1:500; Cell Signaling Technology), anti-DR5, anti-Bid (1:2000; Cell Signaling Technology), anti-β-actin (1:5000; Sigma-Aldrich), anti-caspase-3 (1:2000; R&D, Minneapolis, MN, USA), anti-RNA polymerase II total (RNA Pol II) (1:2000), anti-pSer2 RNA Pol II (1:5000; Covance, Princeton, NJ, USA). Immuno-complexes were detected using peroxidase-conjugated anti-mouse IgG, anti-rabbit IgG, and anti-goat IgG (1:5000; Cell Signaling Technology) followed by chemiluminescence detection (Pierce ECL Western Blotting Solution; Thermo Fisher Scientific).

### 2.8. Statistical Analysis

Data were analyzed using IBM SPSS 21.0 Software (SPSS Inc, Chicago, IL, USA) and GraphPad Prism 5.0 software (GraphPad Software, Inc, La Jolla, CA, USA). Statistical difference between groups was determined using the log-rank test for Kaplan-Meier analysis and Wilcoxon signed-rank test for group comparisons demonstrated by a boxplot. The *p* values < 0.05 were considered statistically significant.

## 3. Results

### 3.1. Patients’ Clinical Characteristics

First, we determined the relevance of CDK9 expression by comparing normal and cancer tissue samples of colorectal cancer patients. The patient characteristics and CDK9 score of our cohort are summarized in Table 1. In this study, a total of 175 patients (93 males and 82 females) with a median age of 69.53 years (range = 29.81–93.98) were analyzed. Of these cohorts, 116 patients were diagnosed with well-differentiated tumors (grade 1 and 2; 67.4%) and 56 with poorly differentiated tumors (grade 3 and 4; 32.6%). Three patients (patients 23, 55, 90) without tumor classification of grading in the database were excluded from the grading analysis. A total of 35 tumors were classified as T1/T2 (20.1%), and 139 tumors as T3/T4 (79.9%). In 90 patients, lymph node metastases were evident (51.7%), but only 52 patients had distant metastasis (29.9%). In summary, 30 patients were assigned to stage I according to the UICC classification (17.2%), and 49 patients were diagnosed with stage II (28.2%). A total of 43 patients belonged to stage III (24.7%), and 52 patients to stage IV (29.9%). Furthermore, the distinction into left- and right-sided colon cancer splits groups into 55 patients with cancer localized in the left colon (31.4%) and 119 patients with cancer grown in the right colon (68%). The median overall survival was 30.57 months ranging from 1.02 to 153.46 months. The 5-year survival was 54.1%. Patient 23 without staging in the database was not included.

### 3.2. High CDK9 Expression in Colorectal Cancer Tissues

In order to investigate the CDK9 expression level in normal and carcinogenic colorectal tissues, we performed immunohistochemical staining using a monoclonal CDK9 antibody on paraffin-embedded adjacent-normal tissue of colorectal cancer patients. We randomly chose 40 colorectal cancer (CRC) patients from our cohort and compared CDK9 expression levels in normal and cancerous colorectal tissues. Representative examples of immunohistochemically CDK9 and hematoxylin-eosin staining (HE) stained human normal and CRC tissues are shown in Figure 1a. Both patients (165 and 89) showed overexpression of CDK9 in cancer tissues (Figure 1a). Strong immunoreactivity could be observed in the cancer tissues of 40 CRC patients (mean: 627.40), revealing a significant difference (*p* < 0.001) compared to normal tissues (Figure 1b). Strong immunoreactivity for CDK9 was also found in five colorectal cancer cell lines (Figure 1c). Thus, a significantly higher CDK9 expression could be detected in colorectal carcinoma tissue compared to normal tissue.

### 3.3. CDK9 as a Positive Prognostic Maker in Colorectal Cancer

Next, we investigated the potential role of CDK9 as a prognostic factor in CRC. We divided the CDK9 immunohistochemical staining scores into two subgroups with a low-expression group with (≤569) and a high-expression group (>569) since CDK9 scores varied from 3 to 1138. A total of 73 patients could be assigned to the first group with a low CDK9 score (mean score = 412) and 102 patients assigned to the high CDK9 score group (mean score = 787) in the carcinoma tissue. Furthermore, 79 patients were deceased at the time point of data collection (04/21/2016).

In order to investigate the potential impact of CDK9 expression on the overall survival of CRC patients, Kaplan–Meier survival estimations were generated by correlating low- and high-expression groups with patients’ clinicopathological parameters listed in Table 2. No significant differences in the survival of the patients could be observed for the total collective, with only a tendency towards longer survival with high CDK9 expression (logrank test: *p* = 0.087; Figure 2a). Representative examples of low- and high-level expressing tumors are shown in Figure 2c.

Interestingly, in the subgroup analysis in females, high CDK9 expression correlated with improved overall survival rates (*p* = 0.011; Appendix A). Furthermore, when subdivided by tumor size, we found a correlation between high CDK9 expression and increased overall survival of CRC patients with T3/T4 tumors (*p* = 0.044; Figure 2b). In contrast, patients with T1/T2 stage revealed no significant correlation between CDK9 expression and overall survival rates. Similarly, we found high CDK9 expression correlated with improved survival in patients with right-sided colon cancer (*p* = 0.037; Appendix A). In summary, CDK9 acts as a positive prognostic parameter at least in a subset of colon cancer patients.

### 3.4. CDK9 Inhibition by Dinaciclib Sensitizes Colorectal Cancer Cell Lines to izTRAIL Treatment

To investigate the sensitivity of colorectal cancer cells to TRAIL, 3-(4,5-Dimethyl-thiazol-2-yl)-2,5-diphenyltetrazolium bromide (MTT) cell viability assays were performed. In this study, the apoptosis-inducing agonist, isoleucine- zipper-tagged TRAIL (izTRAIL), was used to treat five CRC cell lines (HT-29, HCT-116, SW-480, SW-620, and DLD-1). Only the viability of HCT-116 and SW-480 cells was reduced in a dose-dependent matter (Figure 3a). Treatment with Dinaciclib alone inhibited cell viability for all five CRC cells in a time- and dose-dependent manner (Figure 3b). To further evaluate whether CDK9 inhibition is sufficient to re-sensitize colon cancer cells to TRAIL-induced cell death, three TRAIL-resistant CRC cell lines were preincubated with Dinaciclib prior to TRAIL treatment. As shown in Figure 3c, CDK9 inhibition significantly sensitized TRAIL non-sensitive CRC cell lines to TRAIL treatment. Intriguingly, Dinaciclib in combination with TRAIL almost completely obliterated the clonogenic survival of HT-29 cells (Figure 3d). Next, the effect of the combination on the anchorage-independent growth was analyzed by soft agar assays. CDK9 inhibition plus TRAIL suppressed the colony formation potential in comparison to the single treatments in a semi-solid environment (Figure 3e,f). The results shown were replicable in SW-620 and DLD-1 cells (Appendix A). In summary, while single-agent treatment with izTRAIL or CDK9 inhibition by Dinaciclib shows little or no efficacy in suppressing cell growth, the combination treatment is highly effective in inducing cell death.

### 3.5. The Novel Combination with CDK9 Inhibition Enhances TRAIL-Mediated Cell Apoptosis by Apoptotic Pathways

In order to further investigate the mechanism of CDK9-mediated cell death induced by TRAIL, HT-29 cells were treated with izTRAIL, Dinaciclib, and with a combination of both substances. Propidium iodide (PI)-staining and flow cytometry analysis show a significant increase in the sub-G1 group when treated with the combination compared to the single treatment with either Dinaciclib or izTRAIL while overall cell cycle progression was unaffected (Figure 4a,b). Similar results were also found in SW-620 and DLD-1 cell lines (Appendix A). Moreover, cells treated with izTRAIL in the presence of Dinaciclib showed an increase in the cleavage of pro-caspase-8, Bid, pro-caspase-9, and -3 and especially a drastic cleavage of poly adenosine-diphosphate (ADP)-ribose polymerase (PARP) compared to the single treatment in HT-29 cells. (Figure 4c and Appendix A). Taken together, these data demonstrate that CDK9 inhibition enhanced TRAIL-induced apoptosis by the activation of main effector caspases and pro-apoptotic proteins.

### 3.6. CDK9 Inhibition Overcomes TRAIL Resistance by Concomitant Downregulation of the Short-Lived Anti-Apoptotic Proteins Mcl-1 and c-FLIP

To further elucidate the molecular mechanism of apoptosis induction by the combination of CDK9 inhibition and TRAIL, we analyzed the expression of important apoptosis-related proteins. Interestingly, the loss of anti-apoptotic protein Mcl-1 was markedly observed, and the expression of c-FLIP and inhibitor of apoptosis family of proteins (IAPs) was slightly reduced in HT-29 cells, while other proteins remained unchanged (Figure 5a and Appendix A). In order to verify the above results, the effect of Dinaciclib on the expression of the apoptosis-associated proteins CDK9, Mcl-l, and c-FLIP was analyzed in TRAIL-nonsensitive SW-620 and DLD-1 cells (Figure 5b and Appendix A). Overall, we confirmed the previous results. In DLD-1, the reduced expression of c-FLIP was less pronounced, but in both cell lines, the near-complete loss of Mcl-1 is striking. Although Mcl-1 expression of SW-620 showed a stronger expression after 12 h compared to 6 h, expression was nearly lost after 24 h. In these cell lines, CDK9 was degraded time-dependently upon treatment with Dinaciclib.

## 4. Discussion

Colorectal cancer (CRC) remains the second deadliest malignancy for both sexes, especially in Western countries [1,2]. The therapeutic efficiencies of systemic treatment options in CRC are largely limited by their intrinsic or acquired resistance as well as chemotherapy-associated toxicity. The members of the CDK family, as well as their activity regulators, have been regarded as a promising research approach with regard to the pathogenesis and therapy of various diseases, including carcinomas, due to their involvement in many important cellular processes [31]. CDK9 plays a decisive role in the regulation of transcription [32]. Deregulations in this area can consequently contribute to the development of cancer.

The identification of novel drug targets for the treatment of colorectal cancer, preferably with cancer cell specificity is crucial in overcoming treatment resistance. In this study, we demonstrated the overexpression of CDK9 in colorectal cancer tissue compared to normal colon epithelia. An increased expression of CDK9 has already been demonstrated in several types of malignancies, such as pancreatic cancer [18], prostate cancer [33], lymphoma [29], and breast cancer [34]. In addition, CDK9 inhibition was proven effective in ovarian cancer [35], esophageal adenocarcinoma [36], and pancreatic cancer [19].

In the present study, we also demonstrate that, in a subset of patients, CDK9 overexpression served as a beneficial biomarker and was associated with improved survival. Similarly, the high expression of cyclin T1 correlated with an improved disease-free survival of patients with stomach adenocarcinoma [37] and high CDK9 expression was associated with prolonged survival in breast cancer [34]. In contrast, in pancreatic cancer, low CDK9 expression was associated with better survival, especially in patients with well-differentiated carcinomas [18].

The clinical relevance of CDK9 overexpression in colorectal cancer samples is yet to be determined as it is unclear if CDK9 acts as a tumor promotor in colorectal cancer or if the overexpression of CDK9 is a consequence of the cancer genome. The fact that the prognostic value of CDK9 differs between right- and left-sided cancer may support this assumption as both cancer locations differ significantly regarding their genomic background [38]. Thus, further investigations regarding the oncogenic potential of CDK9 in colorectal cancer are warranted.

TRAIL was found to specifically kill cancer cells via apoptosis induction without causing toxicity to benign cells in the 1990s [7,8]. Based on this striking finding, TRAIL receptor agonists (TRAs) have been developed and used in clinical trials. Disappointingly, those trials failed as most of the cancer cells and primary human cancers had developed TRAIL resistance [39]. Much effort has been put into overcoming this resistance [30]. Future TRAIL-based therapies need to be combined with sensitizing agents. The previous study investigated CDK9 inhibition as a promising agent via suppression of c-FLIP and Mcl-1. Thus, we used a novel combination therapy consisting of TRAIL and Dinaciclib, an inhibitor targeting CDK9. Our research demonstrated that the antitumor activity of izTRAIL used in colorectal cancer is substantially enhanced by CDK9 inhibition. It has been demonstrated that CDK9 mediates apoptotic resistance via transcriptionally controlling the overexpression of anti-apoptotic proteins in cancer cells such as c-FLIP and Mcl-1 [15,26]. We showed that Dinaciclib’s activity to sensitize cancer cells to TRAIL-induced apoptosis is due to the downregulation of the beforementioned proteins. The combination therapy of CDK9 and TRAIL may thus represent a potent combination for selectively targeting cancer cells due to the overexpression of CDK9 and the cancer-exclusive expression of TRAIL receptors.

## 5. Conclusions

In summary, we could show that CDK9 is overexpressed in colorectal cancer tissues and CDK9 inhibition facilitates izTRAIL mediated apoptosis via downregulation of the short-lived anti-apoptotic proteins Mcl-l and c-FLIP in colorectal cancer cells. Given its potency, the combination of CDK9 inhibition by Dinaciclib and izTRAIL treatment appears to be an exceptionally promising approach in colorectal cancer therapy. In the age of personalized medicine, especially patients with high levels of CDK9 may profit from this approach.

## Figures and Tables

**Figure 1 biomedicines-11-00928-f001:**
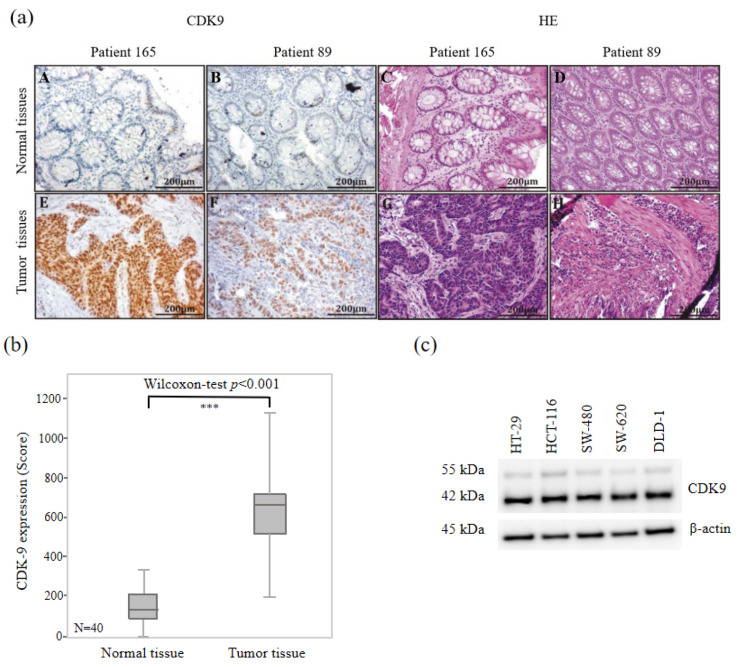
CDK9 expression in normal and colorectal cancer tissue and cell lines. (**a**) Representative pictures of CDK9-Immunohistochemistry and HE staining in human normal and CRC tissue. A,E: CDK-9 IHC staining in normal (A) and tumor tissue (E) of patient 165; B,F: CDK-9 IHC staining in normal (B) and tumor tissue (F) of patient 89; C,G: HE staining in normal (C) and tumor tissue (G) of patient 165; D,H: HE staining in normal (D) and tumor tissue (H) of patient 89 (**b**) Statistical significance between groups was determined using Wilcoxon signed-rank test for group comparison demonstrated by a boxplot. *p* values of <0.05 were considered statistically significant. *** *p* < 0.001. (**c**) CDK9 expression was also shown in different colorectal cancer cell lines. Cells were lysed and subjected to western blotting. One representative of two independent experiments is shown.

**Figure 2 biomedicines-11-00928-f002:**
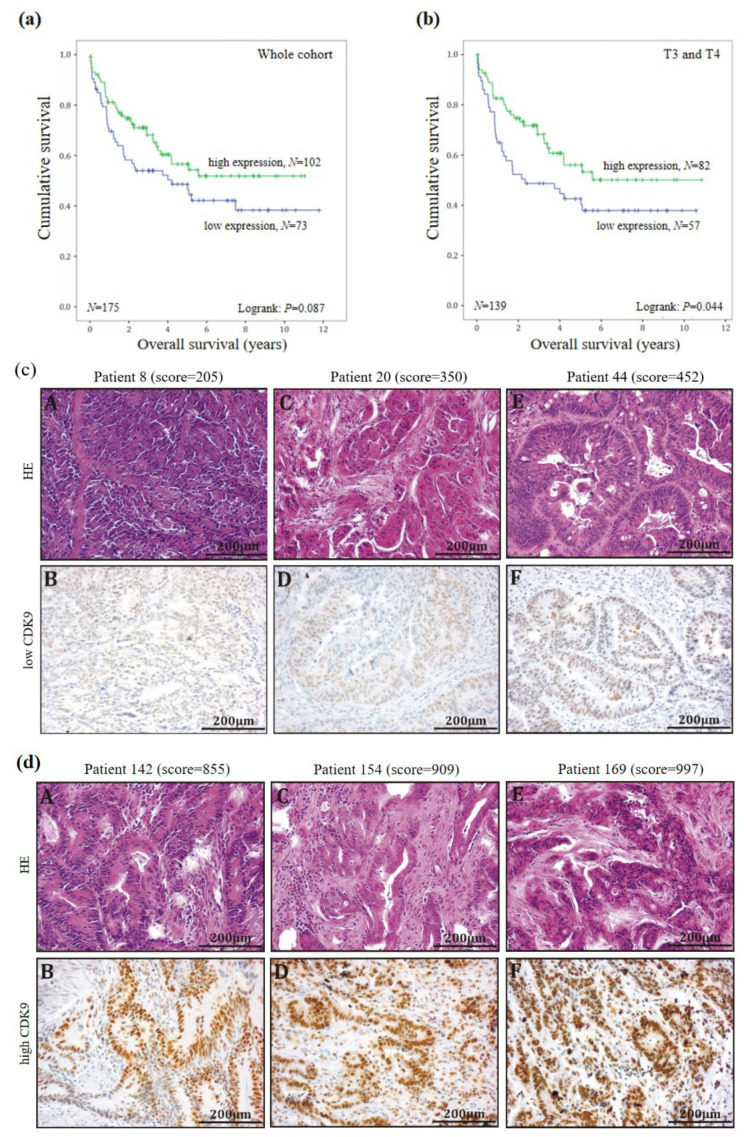
Impact of CDK9 expression on survival of colorectal cancer patients (**a**) and (**b**) Kaplan-Meier survival plot displaying the overall survival of the whole cohort and T3 + T4 according to CDK9 expression in colorectal cancer tissue. Patients with cancer expressing high CDK9 levels show increased survival rates (*p* = 0.087 and *p* = 0.044) compared to those with cancer expressing low CDK9 levels. (**c**) Representative pictures for low CDK9 expressing subgroup (HE and CDK9-immunohistochemistry), A,B: HE and CDK9 expression in patient 8, C,D: HE and CDK9 expression in patient 20; E,F: HE and CDK9 expression in patient 44. (**d**) Representative pictures for high CDK9 expressing subgroup (HE and CDK9-immunohistochemistry), A,B: HE and CDK9 expression in patient 142, C,D: HE and CDK9 expression in patient 154; E,F: HE and CDK9 expression in patient 169.

**Figure 3 biomedicines-11-00928-f003:**
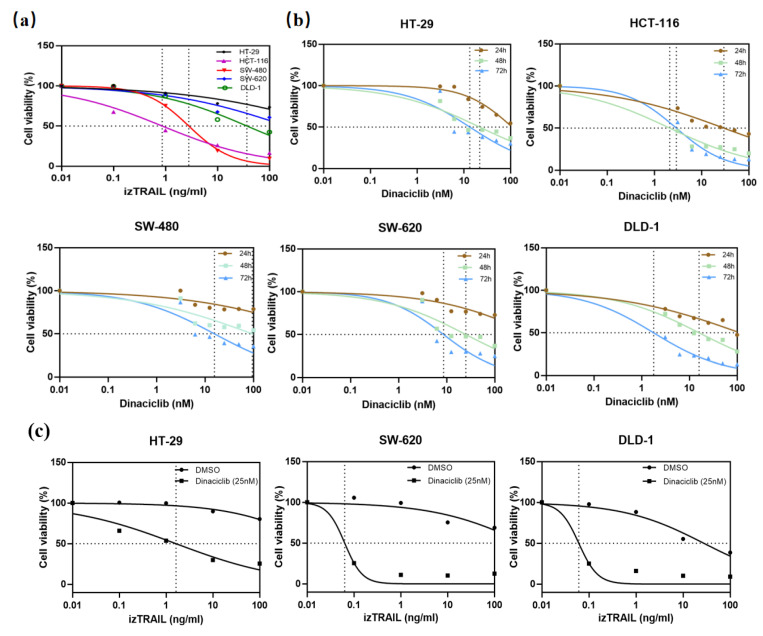
Synergistic effect of Dinaciclib treatment in combination with izTRAIL. Five colon cancer cell lines (HT-29, HCT-116, SW-480, SW-620, and DLD-1) were treated in different ways. (**a**) using izTRAIL (0, 0.1, 1, 10, 100 ng/mL) and (**b**) Dinaciclib (0, 3.125, 6.25, 12.5, 25, 50, 100 nM) for 24, 48, and 72 h. (**c**) preincubated for 4 h with the CDK9 inhibitor Dinaciclib (25 nM) and subsequently treated with izTRAIL at the indicated concentrations (0, 0.1, 1, 10, 100 ng/mL). Cell viability was analyzed by MTT viability assay. Results were normalized to DMSO-treated cells. All values are presented as means ± SD of three independent experiments. (**d**) 50,000 cells/well HT-29 seeded into 6-well plates and were treated with solvent (DMSO) or Dinaciclib (25 nM) for 1 h and subsequently stimulated with izTRAIL for 24 h. Long-term survival was visualized after 7 days by crystal violet staining. (**e**) HT-29 (2000 cells/well) were seeded into 6-well plates in soft agar supplemented with solvent (DMSO), izTRAIL (1, 10 ng/mL), and Dinaciclib (5 nM), respectively, alone or in combination (T1: 1 ng/mL izTRAIL, T10: 10 ng/mL izTRAIL, Combi 1: 1 ng/mL izTRAIL + 5 nM Dinaciclib, Combi 10: 10 ng/mL izTRAIL + 5 nM Dinaciclib). Formed colonies were photographed for three times in a bright field at specific positions located on each fan-shaped center of every well divided into three equal parts and subsequentially visualized and quantified using MTT staining after 21–28 days. One of five independent experiments is shown. (**f**) All values are presented as means ± SD. ** *p* < 0.01, *** *p* < 0.001; Student’s *t*-test.

**Figure 4 biomedicines-11-00928-f004:**
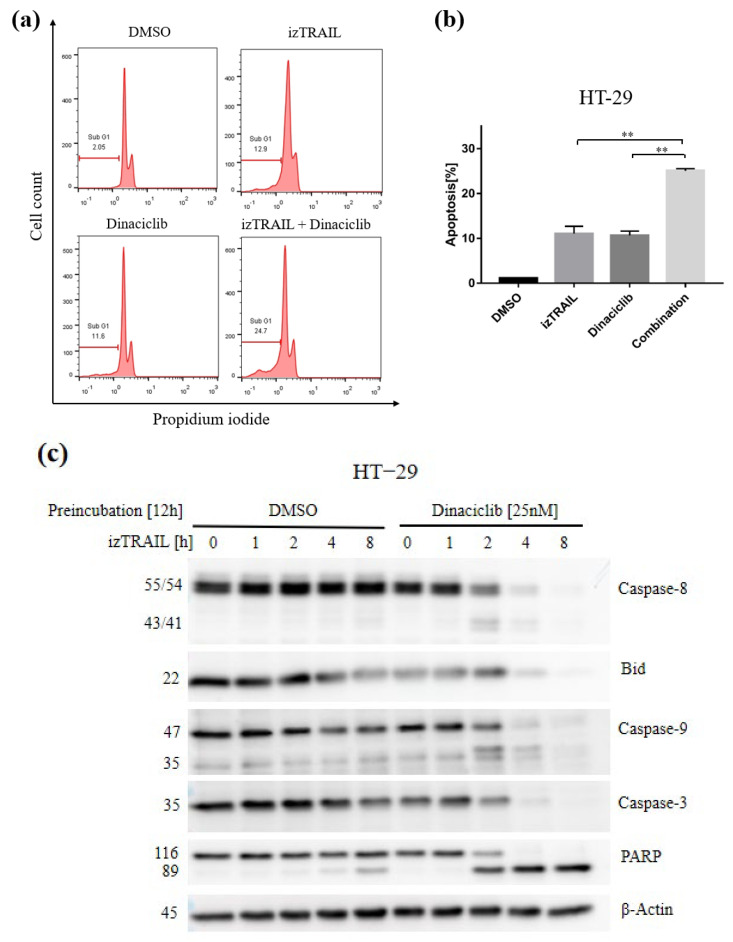
Changes in Sub G1 phase of cell cycle and apoptosis induction in HT−29 cells under the influence of pharmacological CDK9 inhibition plus izTRAIL. (**a**) Analysis of cell cycle Sub G1 phase by propidium iodide (PI) staining for HT−29 cells. Flow cytometry was used to assess Sub G1 DNA content in the entire cell population. Untreated cells (DMSO) were used as negative control; cells were treated either with TRAIL (10 ng/mL) and Dinaciclib (25 nM) or the combination for 24 h. For each treatment condition, 10,000 events were recorded and cellular debris was excluded from the analysis. Data represent one representative of four independent experiments. (**b**) Data are presented as means ± SD; ** *p* < 0.01; Student’s *t*-test. (**c**) HT−29 cells were preincubated with DMSO or Dinaciclib (25 nM) for 12 h and subsequently stimulated with izTRAIL (10 ng/mL) for the indicated periods (0, 1, 2, 4, 8 h). Cells were lysed and subjected to western blotting. Expression levels of Caspase-8, BID, Caspase-9, Caspase-3 and PARP were determinated at the indicated timepoints in control and treated cells. β−actin was used as a loading control. One representative of three independent experiments is shown.

**Figure 5 biomedicines-11-00928-f005:**
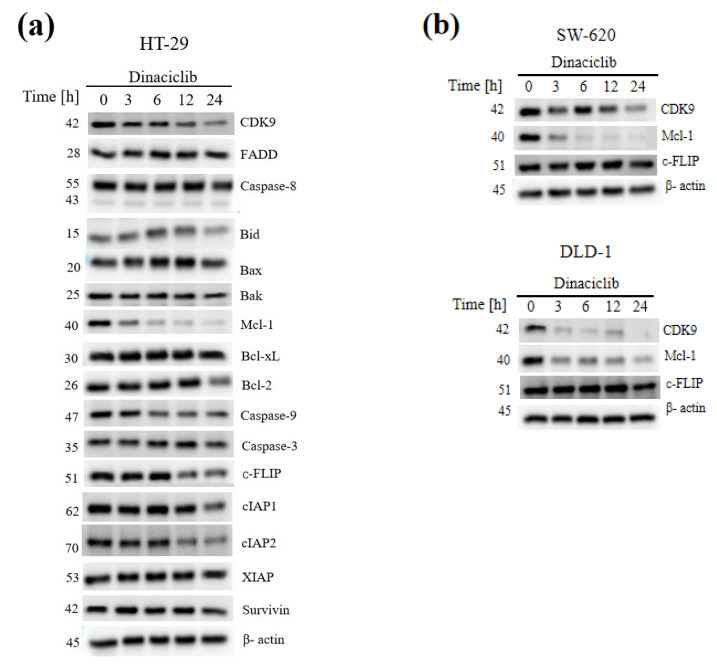
CDK9 inhibition may overcome TRAIL resistance by suppression of c-FLIP and Mcl-1. (**a**) HT-29 cells were treated with 25 nM Dinaciclib for the indicated periods. Cells were lysed and subjected to western blotting to dermine the levels of CDK9, FADD, Caspase-8, Bid, Bax, Bak, Mcl-1, Bcl-xL, Bcl-2, Caspase-9, Caspase-3, c-FLIP, cIAP1, cIAP2, XIAP, and Survivin at the indicated timepoints. β-actin was used as a loading control. (**b**) SW-620 and DLD-1 cells were lysed and analyzed by western blotting concering the expression levels of CDK9, Mcl-1, c-FLIP. β-actin was used as a loading control. One representative of two independent experiments is shown.

**Table 1 biomedicines-11-00928-t001:** CRC patients’ characteristics.

Variable		*N* = 175	%
Gender	Male	93	53.1
	Female	82	46.9
Differentiation	Low (G1 + G2)	116	66.3
	High (G3 + G4)	56	32
	n.d.	3	1.7
T classification	T1, T2	35	20
	T3, T4	139	79.4
	n.d.	1	0.6
LN invasion	N0	84	48
	N1, N2	90	51.4
	n.d.	1	0.6
Distant metastasis	M0	122	69.7
	M1	52	29.7
	n.d.	1	0.6
Stage (UICC)	I	30	17.1
	II	49	28
	III	43	24.6
	IV	52	29.7
	n.d.	1	0.6
Cancer localization	Left colon	55	31.4
	Right colon	119	68
	n.d.	1	0.6
5-year survival			54.1
Age (years)	Mean	69.53	
	Range	29.81–93.98	
Over survival (months)	Mean Range	30.57 1.02–153.46	

Abbreviations: CRC: Colorectal cancer; n.d.: not determined; LN: lymph node; UICC: Union for International Cancer Control.

**Table 2 biomedicines-11-00928-t002:** Correlation of CDK9 expression with clinicopathologic parameters of CRC.

	Total *N*	Low CDK9 Expression [Score ≤ 569] *N* (Median Score)	High CDK9 Expression [Score > 569] *N* (Median Score)	Median CDK9 Expression [Score] (Min–Max)
Follow-up				
Dead	79	40 (394.5)	39 (767)	567 (3–997)
alive	96	33 (450)	63 (796)	643 (143–1138)
Gender				
Male	93	41 (385)	52 (794.5)	618 (3–1094)
Female	82	32 (467.5)	50 (766.5)	623 (143–1138)
T classification				
T1, T2	35	15 (412)	20 (850.5)	651 (6–1080)
T3, T4	139	57 (414)	82 (756.5)	618 (3–1138)
n.d.	1	-		
Localization				
Left colon	55	21 (390)	34 (738.5)	618 (143–1094)
Right colon	119	51 (447)	68 (813.5)	625 (3–1138)
n.d.	1	-		
Stage (UICC)				
I	30	13 (397)	17 (832)	620.5 (6–1080)
II	49	17 (385)	32 (797)	635 (143–1138)
III	43	19 (458)	24 (727)	604 (151–1094)
IV	52	23 (395)	29 (767)	614 (3–1070)
n.d.	1	-		
Total	175	73 (412)	102 (787)	618 (3–1138)

Abbreviations: CDK9: Cyclin Dependent Kinase 9; CRC: Colorectal cancer; n.d.: not determined; UICC: Union for International Cancer Control.

## Data Availability

Not applicable.

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
