# Peer review of "Evaluation of CDK9 Inhibition by Dinaciclib in Combination with Apoptosis Modulating izTRAIL for the Treatment of Colorectal Cancer"

_biomedicines, 2023, doi:10.3390/biomedicines11030928_

Round 1
Reviewer 1 Report
In this manuscript, the authors investigated not only the role of CDK9 in colorectal cancer,but also the potency of CDK9 inhibition as a cancer cell-selective TRAIL-sensitizing strategy. This manuscript is well-organized and clearly stated. I would suggest accepting it after the following concerns are addressed.
1. Please change "apoptosismodulating" to "apoptosis modulating" in the title.
2. Please adjust Table 1 so that it could be on the same page as possible.
3. In figure 3b, horizontal coordinate 10 and 25 overlap.
4. Please add “48 and 72 hours” at the end of "Dinaciclib (0, 3.125, 6.25, 12.5, 25, 50, 100 nM) for 24 hours" in figure note of figure 3b.
5. The dotted lines corresponding to the horizontal coordinate in figure 3a,b are unclear, so it is better to adjust them.
6. It is better to unify the expressions of tumor and cancer in the paper.
7. It is recommended to carry on independent Western blotting experiments at least three times.
8. Please adjust the overlapping words in Figure 5 Coordinate.
9. Please add “the” before “second deadliest” in the first line of the discussion.
10. Please ensure that the citation format of the reference fits the standards for publication.
Reviewer 2 Report
Dear Authors,
I found the revised Manuscript "Evaluation of CDK9 inhibition in combination with apoptosis modulating agents for the treatment of colorectal cancer" (biomedicines-2084672) as good quality considering the level of studies, used methods and the presentation of the results. However, I found few shortcomings which should be improved to consider accepting the manuscript for publication after minor revision. They are as follows:
- Title of manuscript - I find desirable exact indicating CDK9 inhibiting and apoptosis modulating agents used in these studies
- lines: 58, 59, 64, 146, 177, 185 - to much spaces
- line 122 - check the surname “Walcazk”
- 3.2. High CDK9 expression in colorectal tumor tissues – Authors showed the level of CDK9 in normal and tumor tissue sample but only in tumor cell lines. It would be advisable also to show the CDK9 level in normal cell line.
- 3.4. CDK9 inhibition by Dinaciclib sensitizes colorectal cancer cell lines to izTRAIL treatment – figure 3f – in my opinion the necessary combination of statistical analysis are as follows:
T1, Dina vs T1+Dina (combi 1)
T10,Dina vs T10+Dina (combi 10).
Other are redundant and their presentation made the graph analysis difficult.
- 3.5. The novel combination with CDK9 inhibition enhances TRAIL-mediated cell apoptosis by apoptotic pathways - figure 4a – Authors analyzed the results obtained by means of PI staining and flow cytometry only in relation to apoptosis. However, these methods are commonly used also to cell cycle analysis. Have Authors performed this type of analysis?
- figure 5a – the unit of time is unclear (time […]).
Round 2
Reviewer 2 Report
Dear Authors,
thank you very much for taking into account my remarks improving manuscript.
Best regards
